# Effectiveness of a Preventative Program for Groin Pain Syndrome in Elite Youth Soccer Players: A Prospective, Randomized, Controlled, Single-Blind Study

**DOI:** 10.3390/healthcare11172367

**Published:** 2023-08-22

**Authors:** Filippo Cotellessa, Luca Puce, Matteo Formica, Maria Cesarina May, Carlo Trompetto, Marco Perrone, Andrea Bertulessi, Vittorio Anfossi, Roberto Modenesi, Lucio Marinelli, Nicola Luigi Bragazzi, Laura Mori

**Affiliations:** 1Department of Neuroscience, Rehabilitation, Ophthalmology, Genetics, Maternal and Child Health, University of Genoa, 16132 Genoa, Italy; filippo_cotellessa@hotmail.it (F.C.); ctrompetto@neurologia.unige.it (C.T.); perronemarco08@gmail.com (M.P.); andre.bertu@yahoo.it (A.B.); vittorioanfossi@gmail.com (V.A.); robertomodenesi1995@gmail.com (R.M.); lucio.marinelli@unige.it (L.M.); morilaurab@gmail.com (L.M.); 2IRCCS Ospedale Policlinico San Martino, 16132 Genoa, Italy; matteo.formica@unige.it (M.F.); m_may_93@web.de (M.C.M.); 3Orthopedic Clinic, Department of Integrated Surgical and Diagnostic Sciences (DISC), University of Genoa, 16132 Genoa, Italy; 4Laboratory for Industrial and Applied Mathematics (LIAM), Department of Mathematics and Statistics, York University, Toronto, ON M3J 1P3, Canada; robertobragazzi@gmail.com

**Keywords:** groin injuries, soccer, athletes, pain, muscle strength

## Abstract

Groin pain syndrome (GPS) is a prevalent issue in soccer. This study assessed the effectiveness of a new preventive protocol on GPS for youth soccer players. The protocol included targeted stretching and strengthening exercises for the adductor and core muscles from preseason to midseason. A questionnaire and two pain provocation tests were used for the evaluation. Mild GPS required positive results in at least two evaluations, while severe GPS was associated with pain incompatible with engagement in any activity confirmed by diagnostic ultrasound. Forty-two elite male athletes (aged 16.9 ± 0.7 years) participated in the study, with half of them assigned to the usual training (control group) and the remaining athletes undergoing the preventive protocol (treatment group) for 24 weeks. GPS rates were 14.3% (three diagnoses: two mild, one severe) in the treatment group and 28.6% (six diagnoses: three mild, three severe) in the control group. Toward the end of the season, three players, one from the treatment group and two from the control group had to stop playing due to severe GPS problems. In addition, one player in the control group stopped midseason. Even though the reduction in the risk of developing GPS was not significant (relative risk of 0.50 ([95%CI 0.14 to 1.74], *p* = 0.2759), the halved incidence of severe GPS and the increased muscle strength related to the treatment (*p* = 0.0277) are encouraging data for future studies.

## 1. Introduction

Groin pain syndrome (GPS) is characterized by tenderness, functional pain, and sometimes pain affecting the inguinal area, and from the groin, radiating to the medial thigh (insertion of the adductor muscles) or lower abdomen [1]. GPS is also known as athletic pubalgia, inguinal-related pain, sportsman’s hernia, posterior wall weakness, athletic groin pain, or simply groin pain [2]. Although these terms are not completely synonymous and some of them may even be misleading [3], GPS is a widely diagnosed, complex medical issue among athletes, either professional or recreational/amateur [4,5].

The risk factors for GPS, predominantly adductor-related [6,7,8], encompass a significant decline in adductor muscle strength and hip mobility. Engebretsen et al. [9] reported that athletes who exhibit adductor muscle weakness are at nearly four times the risk of developing GPS, in contrast to those who exhibit typical strength levels commonly observed in athletes without any weakness. While GPS can manifest across different sports, certain disciplines pose a higher risk due to the involvement of specific physical stressors. These include, for example, multidirectional team sports, characterized by repetitive and quickly accelerating or decelerating movements, including abrupt changes of direction, and kicking actions [10], such as ice hockey, soccer [11,12,13], and football codes where field kicking is commonly practiced (i.e., Australian football, Gaelic football) [14,15]. In these sports, according to some recent epidemiological surveys, GPS constitutes 7.8% to 12% of all injuries [16,17]. GPS can negatively affect both player and team performance, reducing career longevity, and in some cases, being a career-ending injury [18].

Since previous groin injuries represent a strong risk factor for future groin injuries [18], the need for primary prevention becomes evident. A recent systematic review analyzing the effectiveness of targeted groin injury prevention program [19] found a 19% reduction, which was considered “clinically significant”. This means that although a statistical proof of a reduction in sports-related groin injuries has not yet been achieved [20], the potential impact on athletes’ quality of life and general functioning remains worthy of consideration. Moreover, most research has generally focused on groin pain treatment and management [21,22,23], generally with low-quality studies [24], overlooking the preventative aspect.

Soccer demonstrates the highest prevalence of seasonal groin pain and the longest time loss among various sports, according to a study examining 506 athletes across major team sports [25]. Nevertheless, there is limited research on preventive programs targeting the onset of GPS in soccer players, yielding conflicting outcomes [26,27]. Hölmich et al. [26] conducted a cluster-randomized trial with 1211 players at subelite level from 55 soccer clubs, randomly assigned to the experimental (*n* = 27) or control group (*n* = 28) training as usual. Although a 31% reduction in the risk of groin injury was observed, it was not statistically significant. Conversely, Harøy and colleagues [27] recruited 35 semiprofessional Norwegian soccer teams, dividing them into an intervention (*n* = 18 teams, totaling 339 players) and a control group (*n* = 17 teams, totaling 313 players). A significant reduction of 41% in groin injury risk was calculated (13.5% versus 21.3%) (*p* = 0.008). These conflicting outcomes may be attributed to variations in athlete populations, including age, experience level, and differences in the volume and intensity of the training interventions.

Given both the paucity of data and the scarcity of evidence, further research in the field of groin pain prevention in soccer is urgently warranted. Based on documented risk factors in the existing literature [7,9], we developed an innovative preventive protocol for elite young soccer players without groin pain. This 24-week program is designed to commence during the preseason and conclude midway through the competitive season. It incorporates a combination of stretching and strength exercises, encompassing eccentric, concentric, and isometric muscle contractions, with a progressive load targeting the adductor and core muscles. The primary aim of this randomized, single-blind, controlled trial was to evaluate the effectiveness of the proposed preventive protocol in reducing groin injuries in young soccer players. This evaluation included a comparison of hip and groin pain issues and injuries reported during the study period between participants who followed the preventive regimen in the treatment group and those who did not belong to the control group. A secondary aim was to evaluate the incidence of groin pain among the players enrolled in the study.

Considering the frequency of injuries in the groin area and the presence of specific risk factors, implementing a prevention program that strengthens the adductor muscles and improves hip flexibility could help stabilize the groin area and reduce excessive stress on the tissues; thus, we hypothesized a reduction in the possibility of injury.

## 2. Materials and Methods

### 2.1. Study Protocol and Study Design

The study protocol was reviewed and approved by the local ethical committee of the University of Genoa, Genoa, Italy (protocol number 2022.57). Participation in the study was contingent on obtaining the consent of the participant and his or her parent or guardian. This essential procedural step ensured that all parties involved were well informed and in agreement about the individual’s involvement in the research. The study is registered at the ClinicalTrials.gov public website (identifier: NCT05713487). A prospective, randomized, controlled, single-blind study was conducted. The study included the evaluation of two groups of subjects: an experimental group where the protocol of preventative treatment was implemented two sessions per week, before the usual training session, and a control group that only underwent the usual training. Results are reported in accordance with the “Consolidated Standards of Reporting Trials” 2010 (CONSORT 2010) guidelines [28].

Inclusion criteria were the following: subjects aged fifteen to eighteen years old, competing in the youth sector of a professional soccer club with at least 6 years of training experience. Exclusion criteria included the presence of pain or injuries at the level of the hip/hip region at the initial assessment.

Subjects were randomly allocated to one of the two groups by performing a simple randomization with computer-generated number lists. Single blinding was guaranteed because the investigator who evaluated the athletes did not know to whom the preventive treatment protocol was applied.

A physical therapist, previously trained in the protocol execution and with 15 years of experience in injury prevention and rehabilitation of professional sports, was assigned the responsibility of ensuring accurate and consistent execution of the exercises. All data were collected by two PhD researchers in sports science during the 2021–2022 season, for a total of 42 weeks (6 weeks of preseason and 36 weeks of the season, while the preventive protocol started concurrently with the preseason and consisted of 24 weeks). Standardized questionnaires, pain provocation tests, and diagnostic ultrasound were employed. The clinical evaluation was conducted both before the initiation of the preventive protocol (T0) and at the conclusion of the competitive season (T1). If injuries occurred during this period, athletes discontinued their participation in the study, and the evaluation at T1 was anticipated.

Both the preventive program and clinical evaluation were carried out at the gymnasium (size dimensions: 10 m in length, 15 m in width, and 3.15 m in height; temperature: ≈22°; humidity: ≈50%) of the physical rehabilitation medicine center of Genoa Hospital, Italy located a short distance from the playing field.

### 2.2. Clinical Assessment

Outcome measures were used in a proactive manner to carefully select participants, with a focus on excluding individuals with pain or injury in the hip or hip region (see exclusion criteria). Later in the study, the same measures were used to evaluate the efficacy of the preventive protocol in the selected group of soccer players. A questionnaire was administered to assess patient-reported outcomes (PRO) related to hip and groin pain. In addition, standardized tests were performed to comprehensively assess various aspects, including groin pain, hip joint function, active and passive range of motion, and hip–groin muscle strength. The Hip and Groin Outcome Score (HAGOS) questionnaire [29], which consists of six distinct assessment sections (symptoms, pain, physical function and daily life, functional activities, and sports and recreational activities, performance of physical activities, and quality of life), and comprises 37 items, was employed for the PRO evaluation. For the physical tests, the following assessments were conducted: the Five-Second Squeeze Test (5SST) to evaluate hip and groin function [29], the Flexion-Adduction-Internal Rotation Test (FADIR) to assess hip-joint-related pain [30], the Bent Knee Fall Out Test (BKFO) to measure hip joint mobility [31], and the Hip Adduction Strength (HAS) test to gauge adductor strength [32], with the force measured in newtons using a digital dynamometer (Active Force 2, San Diego, CA, USA). Figure 1 depicts the physical tests used in the clinical evaluation.

Following the completion of the preventive program, each athlete was assigned a score ranging from 0 to 2 based on the severity of their GP syndrome. A score of 0 was assigned if all clinical tests (HAGOS, 5SST, and FADIR) yielded negative results, while a score of 1 (mild GP) was given if at least two tests showed positive results. A score of 2 (severe GPS) was assigned if the groin pain was severe enough to prevent the player from participating in training or the match, necessitating special medical attention [29]. Diagnostic ultrasound was employed to confirm the diagnosis of groin pain in individuals with a score of 2. The examination comprehensively evaluated various anatomical sites, including the adductor origin, abdominal insertion, hip, related structures, pubic symphysis, inguinal canal, and associated soft tissues. This step was taken to ensure an accurate diagnosis and prompt treatment of severe groin injuries. Changes in the structure of the hip tendon and long adductor muscles, as well as thickening and/or lesions in the adductor muscles, were considered. Furthermore, structural or inflammatory abnormalities in the bony structures around the pubic joint and pubic process were also considered as criteria for the diagnosis.

### 2.3. Preventive Program Protocol

The preventive program included five strength exercises and one stretching exercise [26,27], each with a progressive load with the main aim of increasing the strength and joint mobility of the adductor muscles [17]. Before starting the exercise protocol, it was standard practice to perform an individual warm-up lasting approximately 15 min. The exercises and related load parameters of the preventive protocol are described in Table 1, while the execution of the exercises is illustrated in Figure 2.

The warm-up included a series of dynamic exercises, starting with a light jog in place, followed by jumping exercises. Next, joint mobility exercises, such as circular movements involving arms and legs, were performed. Finally, squats and lunges with body weight were included in the routine.

The first type of adduction exercise consisted of isometric hip adduction against a ball placed between the feet under the medial malleoli in the supine position. The second type of adduction exercise involved performing isometric hip adduction against a ball placed between the knees when supine with hips and knees flexed and feet resting on the ground. The third type of adduction exercise was carried out by the player lying on the side of the dominant leg with the dominant limb outstretched and the non-dominant limb hip and knee flexed at 90°. Maximum hip adduction of the dominant leg was performed while keeping the knee straight and the foot horizontal. The fourth type of adduction exercise was conducted by standing with a resistance band at the knee. Finally, the Copenhagen adduction exercise was used as the fifth strength exercise. The sixth and final exercise of the preventative program, known as the seated butterfly exercise, was designed to improve joint mobility to increase the effectiveness of other strength exercises. In that exercise, the player sat on the ground, pressed the soles of their feet together, and used their arms to push their knees towards the floor.

### 2.4. A Priori Sample Size and Power Analysis

An a priori sample size and power analysis for the effectiveness of the preventive treatment was conducted utilizing the following equation [33,34]:(1)$$n=\frac{\chi_{\alpha /2}^{2}\;\cdot \left[\frac{1-P_{1}}{P_{1}}+\frac{1-P_{2}}{P_{2}}\right]}{{\left[ln\left(1-\varepsilon \right)\right]}^{2}}$$
where *α* is the probability of type I error (two-sided significance level) or the probability of rejecting the true null hypothesis, *P*_2_ is the proportion of the nonexposed group (ranging from 9% [35] to 32.5% [25]), *n* is the required sample size for the exposure group, and ε is the relative risk precision (here, it was set at 0.90). *P*_1_ is the proportion of GPS in the exposed group, which can be computed from the following formula:P_1_ = RR∙P_2_(2)
where RR is 0.81 [95%CI 0.60–1.09] [19]. Based on this analysis, N, that is to say, the total sample size needed for the study, was computed to range from 6 to 42 individuals.

### 2.5. Statistical Analysis

Before proceeding with statistical analyses, the data were visually inspected for potential outliers, and the normality of data distribution was verified by applying the Shapiro–Wilk test. This test was preferred over other normality tests given the small sample sizes employed in the current trial. The a priori sample size and power analysis was conducted by means of freely available G*power software v3.1.

Measurements at T0 are presented as mean, standard deviations, and medians (IQR) where appropriate for continuous variables and as absolute frequency (%) for categorical variables, showing overall results and results by groups of treatment. Baseline characteristics of the two groups were compared using the Wilcoxon rank-sum test for continuous variables and Fisher’s exact test for the variable regarding previous injuries. Subsequently, to evaluate the efficacy of the treatment, the two groups were compared based on the occurrence of groin pain (5SST, FADIR, and HAGOS) and on T1–T0 changes in BKFT and HAS test using the Fisher’s exact test or the Wilcoxon rank-sum test as appropriate.

The effectiveness of the protocol in counteracting the insurgence of GPS was estimated by computing the relative risk with its 95% confidence interval, according to Altman [36]. More specifically, the relative risk or risk ratio was computed as:(3)$$RR=\frac{a/(a+b)}{c/(c+d)}$$
where *a* is the number of subjects within the exposed (treatment) group reporting a negative (bad) outcome (i.e., GPS), *b* is the number of subjects within the exposed (treatment) group reporting a positive (good) outcome (i.e., no GPS), *c* is the number of subjects within the control group reporting the negative outcome, and d is the number of subjects within the control group reporting the positive outcome.

All statistical analyses were performed using statistical software SPSS (v.28; IBM, Armonk, NY, USA), and *p*-values < 0.05 were considered statistically significant. A two-sample test of proportions was performed to compare the proportion of cases of groin pain in the two groups.

## 3. Results

Forty-two male athletes aged 16 ± 0.7 years, belonging to a youth academy of a professional soccer club competing in the Italian top league and representing the highest level of youth soccer in Italy, were recruited. Figure 3 illustrates the recruitment and randomization process of the two groups.

No statistically significant differences were found between the treatment and control groups in terms of anthropometric demographic and play characteristics shown in Table 2.

No differences between the two groups could be observed in terms of clinical assessment of pain (5SST, FADIR, and HAGOS), with the only exception of a single HAGOS domain (participation in physical activity). However, the difference was statistically marginal (100 [100 to 100] vs. 100 [87.5 to 100], with a median difference of 0 [0 to 0], *p* = 0.0419) and not clinically meaningful (Table 3).

An increase in hip adduction strength (HAS) could be noted in the treated group and for the left leg (0.2 [0 to 0.4] vs. 0 [−0.2 to 0.2], with a median difference of 0.2 [0 to 0.4], *p* = 0.0277), whilst the effect for the right leg was borderline significant (0.2 [0 to 0.4] vs. 0 [−0.2 to 0.1], with a median difference of 0.2 [0 to 0.3], *p* = 0.0944) (Table 4). Finally, GPS (grade 2) was reported by 9.5% of the recruited athletes.

In the study, one player in the treatment group and three players in the control group were diagnosed with GPS (grade 2). Three of these four players ended the study early due to severe pain. Specifically, one was from the treatment group and two from the control group at week 38, and one from the control group at week 26. The diagnosis in the treatment group was characterized by a bilateral functional overload of the musculature and associated signs of insertional tendinopathy of the short adductor muscle bilaterally. In the control group, inflammatory conditions affecting the adductor tendon insertions on the pubic symphysis, compatible with pubic osteitis, were reported in one subject. In another player, acute grade I distraction injury affecting the proximal myotendinous junction of the long adductor muscle was reported. Finally, in another subject, grade I distraction injury affecting the proximal myotendinous junction of the adductor magnus muscle with only a millimetric discontinuity of muscle fibers was reported, associated with a severe enthesopathy affecting the recto-adductor insertions on the pubic symphysis, with a particular thickening of the common adductor tendon.

The protocol was not effective in counteracting the insurgence of GPS (rate of 14.3%—three diagnoses overall, two of degree 1, and one of degree 2—vs. rate of 28.6%—six diagnoses overall, three of degree 1, and three of degree 2—, *p*_trend_ = 0.5789). This corresponded to a relative risk (RR) of 0.50 ([95%CI 0.14 to 1.74], *p* = 0.2759).

## 4. Discussion

Deficiencies in strength and limited hip mobility contribute to joint instability, compromised movement control, and increased tissue stress, making athletes more susceptible to a higher risk of GPS. Consequently, a meticulously crafted exercise protocol targeting joint strength and mobility in the adductor area was developed by carefully selecting and grouping specific exercises based on their previously proven effectiveness [27]. The present study was devised as a prospective, randomized controlled, single-blind study aimed to investigate the efficacy of a preventive protocol in elite youth soccer players in terms of hip joint and groin pain complaints, as well as GPS incidence. Although the preventive protocol showed a significant reduction in the risk of developing GPS (RR of 0.50), this effect did not reach statistical significance. In addition, the observed incidence of GPS (9.5%) appears to be relatively lower than previously reported in the literature.

### 4.1. Program Effectiveness

While specific exercises have shown some benefit in improving strength, it is possible that this increase may not have effectively targeted the underlying causes or specific risk factors that contributed to the development of pain and injury in the groin area. This could be attributed to the load parameters not adequately stimulating the physical condition of the players. Despite the intervention being progressive, it primarily focused on isometric exercises. Additionally, eccentric contractions were significantly less emphasized in the protocol compared to others [26], and the duration of these contractions during the Copenhagen exercise was limited to only 3 s [37].

In the scholarly literature, only a few studies assessed the effectiveness of preventative programs for GPS, reporting contrasting findings. A systematic review with a meta-analysis of seven studies (six on soccer players) [19] computed an overall effect estimate of the RR of 0.81, ranging from 0.60 to 1.09, with a low heterogeneity (I^2^ = 7%), which was not statistically significant (χ^2^ *p* = 0.37).

When stratifying according to the type of preventive program, two of the trials harnessed “the 11” strategy [38,39], which included core stabilization, thigh muscle eccentric training, proprioceptive training, dynamic stabilization, and plyometrics with straight leg alignment. This preventive program yielded a pooled effect estimate of RR 0.68, spanning from 0.40 to 1.14, with a nonsignificant heterogeneity among studies (I^2^ = 55% χ^2^ *p* = 0.13). Similar results could be obtained with an active adductor muscle strength program, which yielded a pooled effect estimate of RR 0.78, ranging from 0.49 to 1.25, with null heterogeneity among studies (I^2^ = 0%, χ^2^ *p* = 0.34). More specifically, an RCT by Engebretsen and colleagues [9], which recruited 508 players from 31 teams, reported a total of 505 injuries, sustained by 56% of the athletes. The intervention was not effective, potentially due to the low compliance rates (from 19.4% in the groin group up to 29.2% in the knee group).

Contrastingly, the cluster-randomized trial by Hölmich et al. [26] reported a satisfactory compliance rate, even though it still resulted in the program not being effective. The intervention program implemented a preventive protocol consisting of exercises similar to those utilized in the current study; however, it did not include the gradual increase in exercise intensity and volume. The compliance rate was 81% (977 players from 21 clubs in each group out of an initial cohort of 1211 players from 55 soccer clubs). The risk of developing a groin injury was decreased by 31%, which was a nonstatistically significant reduction (with a hazard of 0.69, ranging from 0.40 to 1.19). On the other hand, a recently published study, Harøy and colleagues [27], not included in the above-mentioned meta-analysis, found that an adductor strengthening program was effective in decreasing by 41% the risk of groin injuries, which was statistically significant (*p* = 0.008).

Concerning the determinants and risk factors for GPS, in the present study, there was a borderline significant association between having experienced a previous injury and reporting GPS during the study period. One-third of those who complained of GPS had previous injuries (*p* = 0.1009). This is partially in line with the findings reported by Hölmich et al. [26], who computed that experiencing a previous groin injury approximately doubles the risk of developing a new groin injury. This finding was detected in the univariate but not in the multivariate analysis. Moreover, when excluding subjects with previous injuries from the analysis, the protocol would still be not effective in counteracting the insurgence of GPS. It is crucial to emphasize that in this study, the effectiveness of the preventive protocol was assessed not only based on the occurrence of GPS (GPS grade 2), which refers to a condition that hinders a player from engaging in training sessions or matches until career termination, but also on GPS (grade 1) [5]. Grade 1 GPS is identified by the presence of groin pain, albeit not severe enough to completely halt activities. This methodological decision unavoidably affected the efficacy results of the protocol, but it allowed for a more precise and sensitive evaluation of its application’s impact. However, this methodological choice introduced a potential challenge in terms of comparing the results with those of other studies.

Further discrepancies among studies can be explained, at least partially, considering that GPS represents a complex constellation of diverse, sometimes overlapping, clinical signs and symptoms [4,40], which is a true challenge for the medical sports physician.

### 4.2. Groin Pain Rate

In the present study, severe GPS (grade 2) was reported by 9.5% of the recruited athletes (three players from the control group and one player from the treatment group). All injuries were adductor-related. The severity of the injury was such that it prevented the players from completing the competitive season.

There are many studies that focused on the epidemiology of injuries in the soccer field. Groin injuries represent some of the most common injuries in soccer, accounting for 11% to 16% of all soccer injuries. A 2009 study by Werner and colleagues [6] analyzed the incidence of groin pain in professional soccer players over a seven-year period. The study recorded a total of 628 groin injuries, which accounted for approximately 12–16% of all injuries. Adductor-related groin pain was the most frequent (399 cases), followed by the iliopsoas-related clinical entities (52 cases). Another study conducted by Mosler and colleagues [7] on professional soccer players recorded an incidence of 18% of groin injuries (206 cases of groin pain out of 1145 injuries). Moreover, in this study, it was observed that the most common clinical entities were adductor-related (68%), iliopsoas-related (12%), and pubic-related (9%). Another study by Hölmich and colleagues [8] from 2014, conducted on 998 nonprofessional soccer players, showed that the most common clinical entity of groin pain was adductor-related, followed by iliopsoas-related, and groin-related ones. Our overall rate was lower than those reported in the literature, potentially due to the young age of the sample (16.9 ± 0.7 years). An increasing age has indeed been found to correlate with a higher risk of developing groin injuries [41]. Moreover, the reported percentage in our study accounts for the overall incidence among participants, thereby considering those who underwent the specific GPS prevention protocol. If half of the participants had not adhered to the prevention protocol, the incidence would likely exhibit a lesser deviation from the literature findings.

### 4.3. Strengths and Limitations

Our study has several strengths that should be acknowledged. These include: an elite youth athlete population, the randomized design of the trial, and a thorough clinical assessment during the enrollment of the participants and the completion of the investigation, relying on a standardized terminology and definition of GPS and related reporting. Moreover, a progressive physical activity program was selected (in terms of volume and intensity), thus allowing for a gradual adaptation of the physical exercise to individual needs, thereby reducing the risk of injury and improving the effectiveness of the training. On the other hand, the present trial suffers from some limitations. The major shortcoming is represented by the small sample size employed. However, a smaller number of participants enables almost personalized sessions to be conducted, ensuring that each parameter of the load specified in the preventive protocol is executed perfectly.

Looking forward, there are several areas that deserve attention for future research. First, an analysis of the long-term effects of the prevention program could provide valuable insights into its lasting impact over extended periods. In addition, expanding the study to include a larger cohort of participants without neglecting the accuracy of exercise performance could improve the generalizability of the results and potentially reveal additional nuances.

## 5. Conclusions

GPS represents a common issue in soccer, without effective prevention protocols. Since it has been shown that in athletes with GPS, there is a marked strength reduction in the adductor muscles, we aimed to investigate the effect of a prevention program for GPS in a sample of forty-two elite youth athletes. Even though we found a significant increase in muscle strength in the abductor region of the left leg among participants in the preventive treatment group, our study showed that the protocol targeting adductor muscles halved the risk of developing GPS, but not in a statistically significant fashion. Although not all data were significant, the reduced incidence of severe GPS and the increased muscle strength are encouraging data for future studies. To optimize the effectiveness of GPS preventive protocols, it is advisable to incorporate new exercise programs specifically designed to engage the hip adductors. However, it is equally important not to overlook areas such as hip flexors (e.g., iliopsoas) and rectus abdominis strength to ensure a comprehensive core stability. These findings underscore the need to tailor preventive programs for a more holistic approach to GPS prevention in elite youth athletes.

## Figures and Tables

**Figure 1 healthcare-11-02367-f001:**
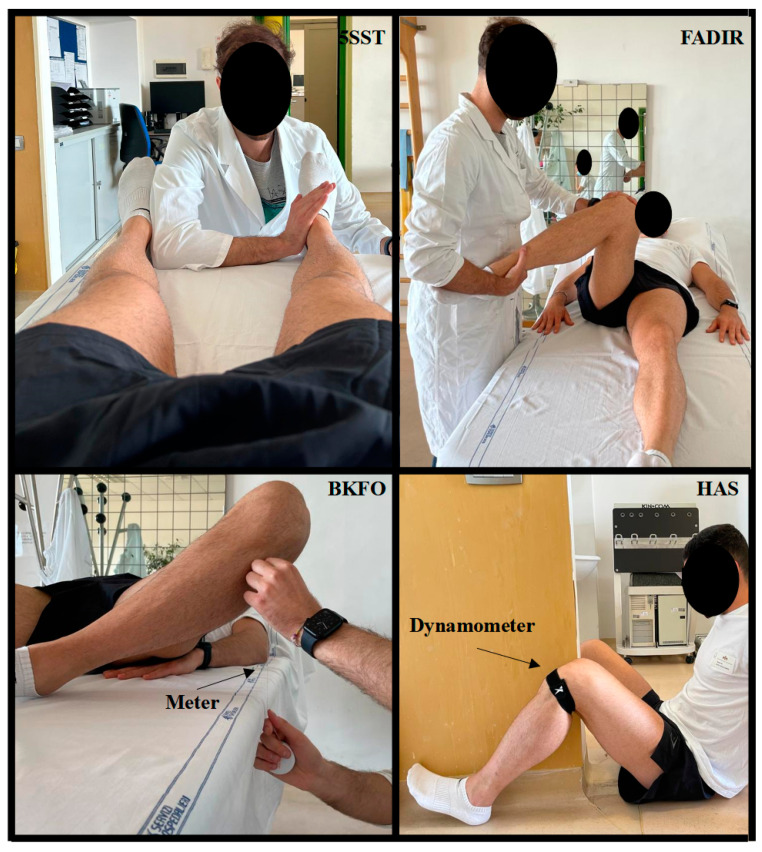
Physical tests used in clinical evaluation. Abbreviations: 5SST, Five-Second Squeeze Test; FADIR, Flexion-Adduction-Internal Rotation Test; BKFO, Bent Knee Fall Out Test; HAS, Hip Adduction Strength.

**Figure 2 healthcare-11-02367-f002:**
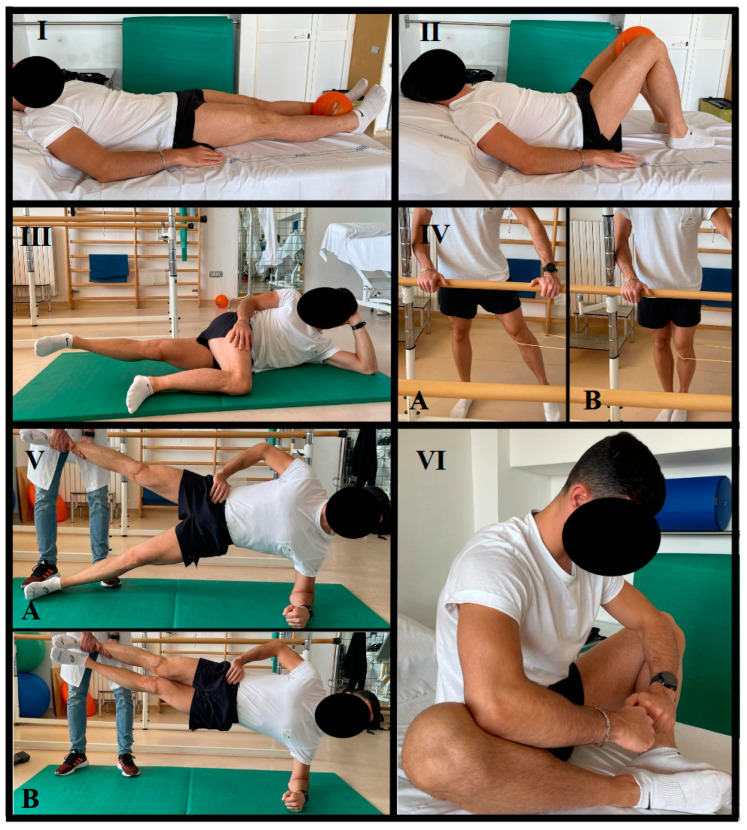
Depiction of the preventive protocol exercises. (I) Isometric adduction with a ball between the ankles. (II) Isometric adduction with a ball between the knees. (III) Lateral hip adduction. (IV) Hip adduction with elastic band. (V) Copenhagen adduction. (VI) Seated butterfly exercise. (A) Starting and ending position; (B) midposition.

**Figure 3 healthcare-11-02367-f003:**
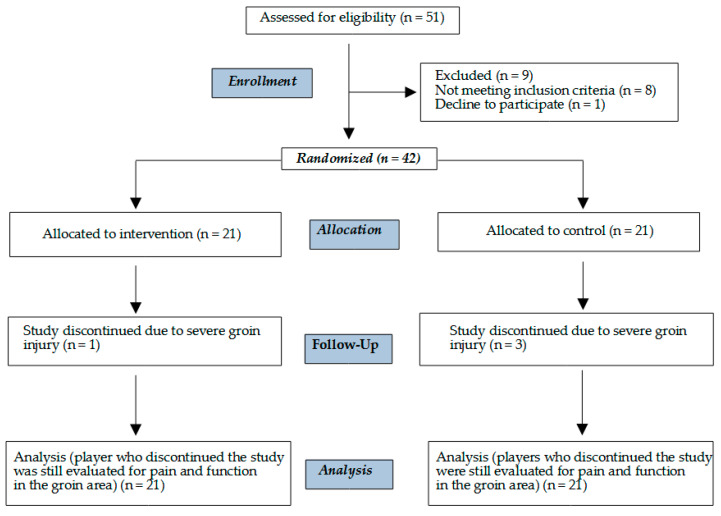
Flowchart of the recruitment and randomization process of the two groups (i.e., enrolment, intervention assignment (allocation), follow-up, and data analysis).

**Table 1 healthcare-11-02367-t001:** The exercises and related load parameters (number of series, number of repetitions, contraction time, relaxation time, and rest time between series) adopted in the preventive protocol.

	1–8th Weeks			9–16th Weeks			17–24th Weeks			
Exercises	Sets	Reps	CT–RT (s)	Sets	Reps	CT-RT (s)	Sets	Reps	CT–RT (s)	Rest (s)
I	1	10	5–5	1	10	10–5	1	10	15–5	/
II	2	8/10	5–5	3	8/10	5–5	4	8/10	5–5	60
III	2	8/10	10–10	3	8/10	10–10	4	8/10	10–10	30
IV	1	5	20–10	1	5	30–15	1	5	60–30	/
V	3	10	6–6	3	15	6–6	3	20	6–6	60
VI	2	5	20–10	3	5	20–10	4	5	20–10	60

I Isometric adduction with a ball between the ankles. II Isometric adduction with a ball between the knees. III Lateral hip adduction. IV Hip adduction with elastic band. V Copenhagen adduction. VI Seated butterfly exercise. Abbreviations: CT, contraction time; RT, relaxation time.

**Table 2 healthcare-11-02367-t002:** Characteristics of the players overall and according to the allocation group.

	Overall (*n* = 42)	Treatment Group (*n* = 21)	Control Group (n = 21)
Age (years)	17.0 ± 0.7	16.8 ± 0.5	17.1 ± 0.8
Height (cm)	178.9 ± 4.9	177.6 ± 4.4	180.1 ± 5.1
Weight (kg)	72.5 ± 5.3	72.3 ± 6.0	72.7 ± 4.5
BMI (kg/m^2^)	22.7 ± 1.3	22.9 ± 1.4	22.4 ± 1.1
Dominant leg (%) (left–right)	19–81	14–86	5–95
Level of play (%) (high–medium)	83–17	86–14	81–19
Training experience (years)	8.7 ± 2.0	9.0 ± 1.9	8.5 ± 2.0
Weekly training load (hours)	10.6 ± 2.3	11.0 ± 2.4	10.1 ± 2.3
Position on field (%)			
Goalkeeper	7	5	10
Defense	33	33	33
Midfield	29	33	24
Striker	31	29	33

Abbreviations: BMI, body mass index. Mean values and standard deviations are reported, unless stated otherwise (percentage).

**Table 3 healthcare-11-02367-t003:** Subjective and clinical outcomes stratified according to treatment and control group and statistical significance.

Parameter	Treatment Group	Control Group	Difference	*p*-Value
HAGOS (overall)	100 [99 to 100]	100 [88.2 to 100]	0 [0 to 1]	0.6642
HAGOS (pain)	100 [100 to 100]	100 [87.5 to 100]	0 [0 to 0]	0.2006
HAGOS (symptoms)	100 [100 to 100]	100 [87.5 to 100]	0 [0 to 0]	0.2008
HAGOS (participation in physical activities)	100 [100 to 100]	100 [87.5 to 100]	0 [0 to 0]	0.0419
HAGOS (functional sports and recreational activities)	100 [100 to 100]	100 [83.6 to 100]	0 [0 to 0]	0.2199
HAGOS (physical function)	100 [100 to 100]	100 [92.5 to 100]	0 [0 to 0]	0.2070
HAGOS (quality of life)	100 [93.8 to 100]	100 [91.6 to 100]	0 [0 to 0]	0.9094
R-5SST	0 [0 to 0]	0 [0 to 2]	0 [0 to 0]	0.2262
L-5SST	0 [0 to 0]	0 [0 to 0.8]	0 [0 to 0]	0.3683
R-FADIR	3 (14.3%)	6 (28.6%)		0.4537
L-FADIR	3 (14.3%)	5 (23.8%)		0.6965
GPS overall GPS grade 1 GPS grade 2	3 (14.3%) 2 (9.5%) 1 (4.8%)	6 (28.6%) 3 (14.3%) 3 (14.3%)		0.4789

Values are medians of the difference between the baseline and the completion of the trial with their IQRs. Abbreviations: HAGOS, Hip and Groin Outcome Score; R-5SST, Copenhagen Five-Second Squeeze Test related to right leg; L-5SST, Copenhagen Five-Second Squeeze Test related to left leg; R-FADIR, Flexion-Adduction-Internal Rotation Test related to right leg; L-FADIR, Flexion-Adduction-Internal Rotation Test related to left leg; GPS, groin pain syndrome.

**Table 4 healthcare-11-02367-t004:** Physiological outcomes stratified according to treatment and control group and statistical significance. Values are medians of the difference between the baseline and the completion of the trial with their IQRs.

Parameter	Treatment Group Delta	Control Group Delta	Difference between Groups	*p*-Value
L-BKFO	−0.1 [−0.2 to 0]	−0.1 [−0.3 to 0]	0 [−0.1 to 0.1]	0.8014
R-BKFO	−0.1 [−0.2 to 0.1]	−0.2 [−0.2 to 0]	0.1 [−0.1 to 0.2]	0.4063
L-HAS	0.2 [0 to 0.4]	0 [−0.2 to 0.2]	0.2 [0 to 0.4]	0.0277
R-HAS	0.2 [0 to 0.4]	0 [−0.2 to 0.1]	0.2 [0 to 0.3]	0.0944

Abbreviations: L-BKFO, Bent Knee Fall Out Test related to left leg; R-BKFO, Bent Knee Fall Out Test related to right leg; L-HAS, Hip Adduction Strength related to left leg; R-HAS, Hip Adduction Strength related to right leg.

## Data Availability

All data generated is within the main text. Further data can be obtained upon request to the corresponding author.

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
