# Peer review of "Effectiveness of a Preventative Program for Groin Pain Syndrome in Elite Youth Soccer Players: A Prospective, Randomized, Controlled, Single-Blind Study"

_healthcare, 2023, doi:10.3390/healthcare11172367_

Round 1

Reviewer 1 Report

I well thought out study which does add value but is limited in it's interventions according to the literature. However, It makes for an interesting read and has clinical implications

L25 – diagnostic ultrasound

L26 – Usual training – add “control group” in brackets

replace “also” with “additionally”

L27 - Preventative protocol – add “treatment group” in brackets

L28-29 – these that stopped playing – were they in the control or treatment group?

L36- “motor pain” = “functional pain”? – probably a better term

L40- “Even” should be a new sentence

L46 – define “normal”

L59-60 – rewrite this as a new sentence, defining “clinically significant”

L70-82 –suggest to differentiate which study had a cohort of professionals and semi-professionals

L98 – as line 25. Also, no need for “closely”

L103 – suggest add and replace “… .stress on the tissues, we hypothesised that there would be a reduction in the possibility of injury”

L111 – suggest add and replace “…where the protocol of preventative treatment was implemented two time…..”

L123 – add if the physiotherapist was taught the protocol beforehand

L106-L128 – mention how consent was obtained – by participant and parent/guardian?

Figure 1 – blank out the persons face in the mirror reflection

L162-163 – early in “participants” you mention who was excluded based on the assessments. Mention here about these groups inclusion/exclusion – I would assume excluded according to the exclusion criteria

L165- as line 25. Mention what was ultrasounded – adductor origin? Abdominal insertion? Hip and related structures?

L173-175 – mention a short description of the warm up

L397-L399 – how was abductor strengthening conclusion drawn? Explain this. What about hip flexors (e.g. ilipsoas) and rectus abdominal strength?

L409-410 – parental/guardian consent?

Some minor english language checks

Author Response

I well thought out study which does add value but is limited in it's interventions according to the literature. However, It makes for an interesting read and has clinical implications

Dear reviewer, thanks a lot for your time spent commenting and reviewing our article. We have taken into full account all your suggestions and we believe we have significantly improved our manuscript. We have performed a high number of revisions, tracked in yellow.

L25 – diagnostic ultrasound

Thanks for the suggestions, “diagnostic” was added: “Mild GPS required positive results in at least two evaluations, while severe GPS was associated with pain incompatible with engagement in any activity confirmed by diagnostic ultrasound”

L26 – Usual training – add “control group” in brackets - replace “also” with “additionally” - L27 - Preventative protocol – add “treatment group” in brackets

Thanks for the suggestions, the sentence has been rephrased like this: “Forty-two elite male athletes (aged 16.9 ± 0.7 years) participated, with half of them assigned to the usual training (control group) and the remaining athletes undergoing the preventive protocol (treatment group)”

L28-29 – these that stopped playing – were they in the control or treatment group?

Thanks for the suggestions, this information has been added: “Toward the end of the season, three players, one from the treatment group and two from the control group had to stop playing due to severe GPS problems. In addition, one player in the control group stopped mid-season”

L36- “motor pain” = “functional pain”? – probably a better term

The term "motor pain" has been replaced by "functional pain," as suggested.

L40- “Even” should be a new sentence

Done, Thank you. Now "also" in the new sentence has been replaced by "although": “Although these terms are not completely synonyms and some of them may be even misleading [3], GPS is a widely diagnosed, complex medical issue among athletes, either professional or recreational/amateur [4,5]”

L46 – define “normal”

In the revised manuscript, the sentence was changed to define "normal": Engebretsen et al. [9] reported that athletes who exhibit adductor muscle weakness are at nearly four times the risk of developing GPS, in contrast to those who exhibit typical strength levels commonly observed in athletes without any weakness.

L59-60 – rewrite this as a new sentence, defining “clinically significant”

The sentence has been changed by better specifying what the authors define as "clinically significant”: “A recent systematic review analysing the effectiveness of targeted groin injury prevention program [19] found a 19% reduction, which was considered 'clinically significant'. This means that, although statistical proof of a reduction in sports-related groin injuries has not yet been achieved [20], the potential impact on athletes' quality of life and general functioning remains worthy of consideration”

L70-82 –suggest to differentiate which study had a cohort of professionals and semi-professionals

We extend our gratitude for your insightful comment, which has not only facilitated the incorporation of a significant detail but has also led to the omission of an inaccurate piece of information. While the Harøy et al. study explicitly designates soccer players as semiprofessionals, the Hölmich et al. study does not. In the latter study, to be specific, the term "sub-elite level" is employed. We have introduced "sub-elite level" in the description of the first study, intending it to be synonymous with "semiprofessional." Subsequently, we have opted to eliminate the distinction between professional and semiprofessional as a potential criterion for discerning heterogeneity of outcomes between the two studies.

In addition, we reduced paragraph 65 86 which was too long.

L98 – as line 25. Also, no need for “closely”

The inclusion of "Diagnostic ultrasound" was extended to all relevant sections of the manuscript. Lines 97-99 were suggested for relocation to the methods section by reviewer 3. However, it is essential to clarify that only the sentence "Standardized questionnaires, pain provocation tests, and diagnostic ultrasound were employed" was indeed relocated. Conversely, the sentence "Athletes were monitored from the initiation of the preventive protocol until the conclusion of the soccer season" was omitted, as it redundantly reiterates information already presented in the methods section.

Precisely, the sentence "Standardized questionnaires, pain provocation tests, and diagnostic ultrasounds were used" was integrated into the subtitle "Protocol and study design."

L103 – suggest add and replace “… .stress on the tissues, we hypothesised that there would be a reduction in the possibility of injury”

We thank you for the suggestion the sentence was rephrased in this way: Considering the frequency of injuries in the groin area and the presence of specific risk factors, implementing a prevention program that strengthens the adductor muscles and improves hip flexibility could help stabilize the groin area and reduce excessive stress on the tissues, we hypothesized a reduction in the possibility of injury.

L111 – suggest add and replace “…where the protocol of preventative treatment was implemented two time…..”

We thank you for the suggestion the sentence was rephrased in this way: The study included the evaluation of two groups of subjects: an experimental group where the protocol of preventative treatment was implemented two time, before the usual training session and a control group that underwent only the usual training.

L123 – add if the physiotherapist was taught the protocol beforehand

Done, Thank you. A physical therapist, previously trained in protocol execution and with 15 years of experience in injury prevention and rehabilitation of professional sports, was assigned the responsibility of ensuring accurate and consistent execution of the exercises.

L106-L128 – mention how consent was obtained – by participant and parent/guardian?

Done, Thank you. Participation in the study was contingent on obtaining the consent of the participant and his or her parent or guardian. This essential procedural step ensured that all parties involved were well informed and in agreement about the individual's involvement in the research.

Figure 1 – blank out the persons face in the mirror reflection

In Figure 1, the identities of all individuals present have been obscured.

L162-163 – early in “participants” you mention who was excluded based on the assessments. Mention here about these groups inclusion/exclusion – I would assume excluded according to the exclusion criteria

We thank the reviewer for this helpful comment. As a result, we recognized that the clarity of the description of our methods needed improvement. Subtitle 2.2, entitled "Participants," already contained an account of the subjects who met the inclusion and exclusion criteria and then participated in the study. To improve the reader's understanding, this section was appropriately moved to the results.

In the subsequent subtitle "2.3. Clinical assessment," we better explained the study design: " Outcome measures were used in a proactive manner to carefully select participants, with a focus on excluding individuals with pain or injury in the hip or hip region (see Exclusion Criteria). Later in the study, the same measures were used to evaluate the efficacy of the preventive protocol in the selected group of soccer players. A questionnaire was administered to assess patient-reported outcomes (PRO) related to hip and groin pain. In addition, standardized tests were performed to comprehensively assess various aspects, including groin pain, hip joint function, active and passive range of motion, and hip-groin muscle strength”.

With these changes, we believe that the reader can now better understand the context of lines 162-163, which pertain only to recruited participants who will undergo the second clinical evaluation at the end of the study.

L165- as line 25. Mention what was ultrasounded – adductor origin? Abdominal insertion? Hip and related structures?

"Diagnostic" has been added plus suggested information has been included: “Diagnostic ultrasound was employed to confirm the diagnosis of groin pain in individuals with a score of 2. The examination comprehensively evaluated various anatomical sites, including the adductor origin, abdominal insertion, hip, related structures, pubic symphysis, inguinal canal, and associated soft tissues.”

L173-175 – mention a short description of the warm up

It was a short description of the warm-up as suggested: “The warm-up included a series of dynamic exercises, starting with a light jog in place, followed by jumping exercises. Next, joint mobility exercises, such as circular movements involving arms and legs, were performed. Finally, squats and lunges with body weight were included in the routine”

L397-L399 – how was abductor strengthening conclusion drawn? Explain this. What about hip flexors (e.g. ilipsoas) and rectus abdominal strength?

First of all, we apologize because there is an error in the text (lines 398-399) it talks about adductor muscles and not abductor muscles. This error has been corrected in the revised manuscript. Furthermore, the reinforcement of adductor strength, as elucidated in the conclusion, stands out as a noteworthy outcome within the cohort that underwent the preventive treatment—a group that also exhibited a lower incidence of both mild and severe groin pain. Given that a strength deficit in this area constitutes a risk factor, its enhancement through strengthening exercises holds the potential to mitigate the risk of injury.

However, we realize and again thank the reviewer for this that concluding the study with this suggestion might be misleading. We modify lines 397-399 to read as follows:

" To optimize the effectiveness of GPS prevention protocols, it is advisable to incorporate new exercise programs specifically designed to engage the hip adductors. However, it is equally important not to overlook areas such as hip flexors (e.g., iliopsoas) and rectus abdominis strength to ensure comprehensive core stability. These findings underscore the need to tailor preventive programs for a more holistic approach to GPS prevention in elite youth athletes”.

L409-410 – parental/guardian consent?

Informed consent was obtained from all subjects involved in the study and their parents or guardians. This part was added to the revised manuscript.

Reviewer 2 Report

This study is a RCT to examine the effectiveness of a preventative program for GPS.

This article needs to be edited to make it concise and easy for readers to read.

Author Response

This study is a RCT to examine the effectiveness of a preventative program for GPS.

This article needs to be edited to make it concise and easy for readers to read.

Dear reviewer, we extend our sincere gratitude for your dedicated time and effort invested in providing valuable feedback and reviewing our article. We also appreciate the contributions of the other three reviewers who, like you, pointed out the need to make some specific sentences or concepts more concise and easier for the reader. In response to your collective insights, we have worked to enhance the quality and clarity of our study. Incorporating your suggestions and those of your esteemed colleagues, we believe that our article now offers a more comprehensive and coherent representation of our research findings. For your convenience, all revisions have been meticulously documented in yellow within the revised manuscript.

Reviewer 3 Report

Dear authors,

Please review the document.

Thank you.

Author Response

Please look at the attachment, thank you

Reviewer 4 Report

Dear Authors,

I would like to express my gratitude for the opportunity to review this manuscript.

At this stage, the document requires improvements, below with line indication:

5-7 – In some author’s space and others not between name and affiliation number. Please revise.

8-19 – Please revise the affiliations. Journal normally requires zip codes.

65-86 – The paragraph is too long, please consider standardizing 8-12 lines to improve readability.

87-103 – Please consider more precisely indicating the aim of the study, preferably in the last line of this paragraph, to provide readers a clear and direct information.

107-110 – Informed consent filled by guardians?

130-134 – Please include all details related to the sample. Players’ level, experience, training routines, and others.

136 – Please revise the table content. For example, in “15.8 ± 0.5” more than one space.

135 – The abbreviations should be placed in the table footnote. Same in lines 179 (table 2), 254-257 (table 3), and 262-264 (table 4).

139 - Please describe the procedures in detail. Time of data collection, local characteristics (equipment, size, temperature, humidity, and others). Moreover, who collected the data? Academic background, experience? All these and other details are very important to be clearly understood by readers.

154 – It is suggested that the face of all the involved persons in the figure is not shown.

172 – What is the literature background to support the preventive program protocol? Please describe with references.

190-210 – Please consider splitting the paragraph (too long).

213 – “[33, 34]”, please remove the space.

233 – “(v.28; IBM)”, please include city and country.

224-235 – Please describe all procedures in detail, for example, normality test.

247 – “Newton”, please revise, looks like a mistake.

260 – Please consider including text between tables.

284 – Discussion section – Please present the aim of the study and the main findings, comparing afterward with the literature in the discussion section.

325 – “1,211” This format is not according to the manuscript content.

305-350 – The paragraph is too long, consider 8-12 lines for each paragraph in all manuscript. The discussion section should globally improve, with more references (very few new references compared to previous sections).

387 – Please describe suggestions for future research.

389-399 – Please consider paragraphs to provide clear and direct messages.

400-411 – Some topics are missing, for example, funding. Please consider the journal template.

412 – Please revise the references format. Some examples. Titles in upper and lowercase; journals in full or abbreviated. Please check all details.

 Moderate editing of English language required.

Author Response

(The authors gave the same response as above.)

Round 2

Reviewer 1 Report

Thank for these acceptable changes to the manuscript

Author Response

We are happy that our work was comprehensive.
Thank you again. 

Reviewer 3 Report

Dear Authors,

Thank you for your responses to corrections and suggestions.

Greetings.

Author Response

(The authors gave the same response as above.)

Reviewer 4 Report

Dear Authors,

Thank you for considering my suggestions and incorporating them into the manuscript, which is globally improved, congratulations.

Below are suggestions related to this last version (v2), with line indication.

Please correct the header in all document “Healthcare 2021, 9, x FOR PEER REVIEW”

26 – “participated” – “participated in the study” – Suggested.

27 – Please indicate the time frame of the intervention period, it is an important information inan abstract.

109 – Not clear the new information in yellow “two time”, two sessions a week? Please provide more precise information.

132 – “10x15m h3.15m” please revise the text aiming at readers interpretation.

180 – Some weeks with “-“ in the middle, others not. Please standardize.

187 – The figure does not mention “Abbreviations: CT, contraction time; RT, relaxation time”.  Please revise.

227 – Please include sample power data (Gpower).

249 – Please remove the line.

261 – In the first column only variables and units should be presented. The “M ± SD” information should be provided in the table title and footnote in full.

266 “p” should be presented in italic in all document. Please revise.

261-270 – Table 2 text is aligned at left, table 3 text centered. Please standardize the format in all document (all tables).

271 – Text above the line. Please revise.

283-285 – Some evaluations name in uppercase, and others in lowercase. Please standardize.

341 – “Hölmich et al. [27]” is indicated, although, the reference number is 26 at the end of the manuscript. Please carefully revise all the citation numbers and correspondent references.

430 – “Data Availability Statement” is missing.

444 – Please carefully revise all references format. For example, the ref 20 journal in V2 is incorrectly in full.

Please revise the English and Manuscript format considering the journal template and instructions for authors.

Minor editing of English language required.

Author Response

Thank you for your valuable suggestions. All typos have been corrected and all suggestions have been changed as requested. All changes are underlined in yellow.